# Conformational Properties of New Thiosemicarbazone and Thiocarbohydrazone Derivatives and Their Possible Targets

**DOI:** 10.3390/molecules27082537

**Published:** 2022-04-14

**Authors:** Nikitas Georgiou, Aikaterini Katsogiannou, Dimitrios Skourtis, Hermis Iatrou, Demeter Tzeli, Stamatia Vassiliou, Uroš Javornik, Janez Plavec, Thomas Mavromoustakos

**Affiliations:** 1Laboratory of Organic Chemistry, Department of Chemistry, National and Kapodistrian University of Athens, Panepistimioupolis Zografou, 11571 Athens, Greece; nikitage@chem.uoa.gr (N.G.); katerinakatsogiannou10@hotmail.com (A.K.); 2Laboratory of Polymer Chemistry, Department of Chemistry, National and Kapodistrian Nikitas Georgiou University of Athens, Panepistimioupolis Zografou, 11571 Athens, Greece; skourtisd@chem.uoa.gr (D.S.); iatrou@chem.uoa.gr (H.I.); 3Laboratory of Physical Chemistry, Department of Chemistry, National and Kapodistrian University of Athens, Panepistimioupolis Zografou, 11571 Athens, Greece; tzeli@chem.uoa.gr; 4Theoretical and Physical Chemistry Institute, National Hellenic Research Foundation, 48 Vassileos Constantinou Ave., 11635 Athens, Greece; 5Slovenian NMR Centre, National Institute of Chemistry, SI-1001 Ljubljana, Slovenia; uros.javornik@ki.si (U.J.); janez.plavec@ki.si (J.P.)

**Keywords:** thiosemicarbazones, thiocarbohydrazones, NMR spectroscopy, quantum mechanics, molecular binding, DFT

## Abstract

The structure assignment and conformational analysis of thiosemicarbazone **KKI15** and thiocarbohydrazone **KKI18** were performed through homonuclear and heteronuclear 2D Nuclear Magnetic Resonance (NMR) spectroscopy (2D-COSY, 2D-NOESY, 2D-HSQC, and 2D-HMBC) and quantum mechanics (QM) calculations using Functional Density Theory (DFT). After the structure identification of the compounds, various conformations of the two compounds were calculated using DFT. The two molecules showed the most energy-favorable values when their two double bonds adopted the *E* configuration. These configurations were compatible with the spatial correlations observed in the 2D-NOESY spectrum. In addition, due to the various isomers that occurred, the energy of the transition states from one isomer to another was calculated. Finally, molecular binding experiments were performed to detect potential targets for **KKI15** and **KKI18** derived from SwissAdme. In silico molecular binding experiments showed favorable binding energy values for all four enzymes studied. The strongest binding energy was observed in the enzyme butyrylcholinesterase. ADMET calculations using the preADMET and pKCSm software showed that the two molecules appear as possible drug leads.

## 1. Introduction

The framework 1,3-diphenylprop-2-en-1-one (Figure 1A) is well known by the generic term ‘‘chalcone’’, a name coined by Kostanecki and Tambor [1]. Chalcones belong to the flavonoid family, and they contain conjugated double bonds with absolute delocalization and two aromatic rings. They act as synthons by which a range of analogs and novel heterocycles with pharmaceutical structures can be targeted [2]. Chalcones can be used to obtain several heterocyclic rings through ring closure reactions [3]. Various chalcone derivatives show antimicrobial [4], antifungal [5], antimalarial [6], antiviral [7], anti-inflammatory [8], antileishmanial [9] anti-tumor, and anticancer [10] properties. 

Thiosemicarbazones (TCSs) (Figure 1B) are an important class of compounds possessing remarkable biological properties making them of interest to structural and medicinal chemists. A wide variety of TSCs containing an appropriate structural framework were found to have antineoplastic [11], antibacterial [12], and antifungal [13] properties. Furthermore, their extraordinary complexing capacity with metal ions such as iron, copper, and zinc provides additional versatility as potential candidates for the preparation of coordinate complexes [14]. Thiosemicarbazones are well known to exhibit both *syn* and *anti* isomeric forms [15].

In recent years, there has been growing interest in the coordination chemistry of thiocarbohydrazones, compounds that share the general formula depicted in Figure 1C and that can be considered the higher homologues of thiosemicarbazones. The first synthesis of these systems is dated in 1925 and described the condensation of ketones and aldehydes with thiocarbohydrazide [16,17].

Compounds combining these structural features (**KKI15**: chalcone-thiosemicarbazone) (**KKI18**: chalcone-thiocarbohydrazone) have never been studied, and the elucidation of their structure is of utmost interest in the understanding of their biological results. Finally, these compounds have great potential for research activities as they are potential lead drugs for diseases to be discovered.

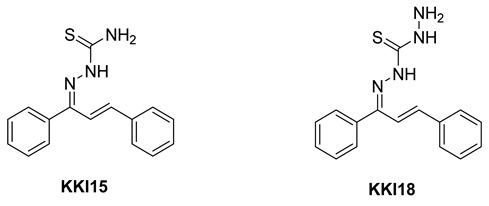


## 2. Results and Discussion

### 2.1. Structure Assignment

As a convenient starting point for the structure assignment of **KKI15**, the readily assigned H-7, which resonates at 6.79 and 7.20 ppm, was used. These two signals are due to the two conformations of **KKI15**. Through 2D-COSY, H-6 resonates at 6.46 ppm. Through 2D HSQC, the H-7 and H-6 show ^1^J_C-H_ coupling with the C7 and C6, respectively, and therefore, C7 and C6 are assigned unambiguously at 118.0 and 143.0 ppm, respectively. Through 2D-NOESY, H-9 is identified at 7.83 and 8.31 ppm due to its correlation with H-7. H-10 and H-11 are then identified due to their correlation with H-9, through 2D-COSY. Furthermore, it is observed that H-7 has two more correlations with protons H-10 and H-13. Through 2D-HSQC, H-9, H-10, and H-11 show ^1^J_C-H_ coupling with C-9, C-10, and C-11, respectively, and, therefore, C-9, C-10, and C-11 are assigned unambiguously at 119.10, 129.37, and 129.65 ppm. Through 2D-COSY, H-13, H-14, and H-15 are identified, and through 2D-HSQC, C-13, C-14, and C-15 are therefore assigned unambiguously at 127.39, 128.76, and 131.27 ppm. Moreover, it is observed in one correlation between H-6 and H-14. Protons attached to nitrogen are increasingly deshielded. NH(3) resonates at 11.10 ppm. Through 2D-NOESY, NH_2_ resonates at 7.80 ppm. Through a two-dimensional ^13^C-^1^H spectrum of the **KKI15**, all the carbons were identified except for the quaternary and carbonyl carbons. These carbons have been identified through 2D-HMBC. Specifically, H-7 shows ^3^J_C-H_ with C-5, H-6 shows ^3^J_C-H_ with C-8, H-13 shows ^2^J_C-H_ with C-12, and finally, NH shows ^2^J_C-H_ with C-2. Based on this strategy, the complete identification of all the proton and carbon atoms of the **KKI15** molecule was achieved.

A similar procedure was carried out for thiocarbohydrazone KKI18. Two signals were observed for H-8 due to the two conformations of **KKI18**. The only difference is that **KKI18** has one more amine group than **KKI15**. For that reason, NH(4) is resonated to 11 and 9.62 ppm, while through 2D-NOESY, NH2 is identified at 3.33 and 5.02 ppm. Finally, through 2D-NOESY, NH(2) is identified as it is associated with NH2. The NH makes sense to be increasingly deshielded because it is next to two electronegative individuals. Moreover, there were observed correlations similar to **KKI15** between H-8 and H-10, H-11 and H-14, and H-7 with H-15.

All the calculations were carried out in DMSO because it is considered to be a solvent that simulates the amphoteric environment and is suitable for the observation of NOE effects [18,19].

The two identification strategies are shown in Appendix A. The two proton spectra of **KKI15** (Figure 2 and Figure 3) and **KKI18** (Figure 2 and Figure 3) are shown below. Moreover, Table 1 with the chemical shifts of these two compounds is shown below.

### 2.2. Conformational Analysis

DFT was used to predict the lowest energy conformations for **KKI15** and **KKI18** as this method offers the highest accuracy of the existing ones. Various initial structures were geometry optimized, and all geometry calculations resulted in eight conformers for **KKI15** and for **KKI18**, see Figure 4. Then their frequencies were calculated. No imaginary frequencies have been found confirming that there are true minima structures.

Four dihedral angles were selected for each molecular structure. Specifically, the angles of **KKI15** that were selected for DFT are formed by the following atoms: 5-6-7-8 (τ1), 3-4-5-6 (τ2), 3-4-5-12 (τ3), and 8-7-6-5 (τ4) and for **KKI18**: 6-7-8-9 (τ1′), 4-5-6-7 (τ2′), 4-5-6-13 (τ3′) and 9-8-7-6 (τ4′). The relative energies are given in Table 2 with the values of dihedrals angles for **KKI15** and **KKI18**.

In Table 2, the values of the dihedral angles differ in each isomer. Although the theoretical value of the dihedral angle should be 0° or 180°, there are some small deviations from these values. This may be due to stereochemical hindrances.

Table 2 clearly shows the relationship between the energy values and the structures, with the *E* isomers having lower energies values than the *Z*. From the three-dimensional structures, it is observed that the *E* configurations reduce the stereochemical repulsions in relation to the *Z*. Figure 5 shows the hydrogen bonds between the hydrogen of amines and nitrogens for (a) **KKI15** and (b) **KKI18**. The same hydrogen bonds were observed for all the conformations of each compound.

Considering the predicted energy values, the structures of the compounds, and the correlations that were observed in 2D-NOESY spectra (Figure 6), the structures MS1 and MS5 are taken as the most probable configurations for **KKI15** and **KKI18,** respectively, as shown in Figure 7 (Table 3).

### 2.3. Energy of Transition States

In order to calculate the transition state structures, an initial conformation with dihedral angles τ_1_ or τ_2_ of 90° was used, which subsequently was geometry optimized. For the calculation, the geometries of the isomers were considered as implemented in the STQN methodology. The values of the energy of transition states for **KKI15** are shown in Table 4. Figure 8 and Figure 9 show the diagrams for the *cis-trans* isomerization energy barrier for compound **KKI15**.

The values of the energy of transition states for **KKI18** are shown in Table 5. Figure 10 and Figure 11 show the diagrams for the *cis-trans* isomerization energy barrier for compound **KKI18**.

The first two energy gaps for **KKI15** have very high values (larger than 20 kcal/mol), and for this reason, these conformations are not observed in the NMR spectrum. On the contrary, for **KKI18**, the energy gap between MS5 and MS6 is less than 20 kcal/mol. This explains the experimental result of observing two conformations. Moreover, the stabilization of the transition states due to the explicit inclusion of the solvent has been evaluated. Specifically, the addition of a solvent molecule in the reaction path MS1→TS3→MS2 of **KKI15** results in further stabilization of about 10 kcal/mol, and the energy gap was found to be 53.8 kcal/mol. For the remaining transition states structures, the explicit inclusion of the solvent does not further stabilize them.

### 2.4. Population Calculation

To make the identification of molecules easier, charge calculations were performed in double bond protons C6=C7 and C7=C8 for compounds **KKI15** and **KKI18**, respectively. This will determine which carbon and proton are the most deshielded in the NMR spectrum. For the protons, the calculations showed similar charges. Specifically, CM5 showed a slightly increased positive charge for proton 7 than 6, while the NMO showed 6 to be more positive than 7. On carbons, CM5 showed a more positive charge on C7 by 0.01, while NBO and Mulliken showed carbon 6 to be more negative than 7 by 0.08 and 0.14, respectively. Due to such small differences, a firm conclusion cannot be derived. The charges of carbons range from 0.01 to 0.1, depending on the method. The detailed results are shown in the Appendix A).

In the next step, molecular orbitals were calculated in each conformation. For both compounds, the HOMO molecular orbital is localized in the sulfocarbonyl group, and LOMO molecular orbital is localized in the sulfocarbonyl group and in the aromatic ring next to the double bond. The figures with the molecular orbitals are also shown in Supporting Information.

### 2.5. ECD Results

Absorption UV-Vis spectra of all conformers are given in SI. It was found that there are triplet excitations in the visible area only for **KKI15**, specifically in the region 625–750 nm. Absorption spectra of **KKI18** conformers show no triplet excitation in the visible region. Finally, both compounds present intense single excitations in the UV region at about 230 nm.

### 2.6. Circular Dichroism (CD)

As it is observed from the CD spectra, in KKI15, there is a distinct difference between the two isomers; in contrast with KKI18, where this difference appears very small. That explains the computational results in which KKI18 was found to show a very low energy gap between the two conformations (See Appendix A).

### 2.7. Molecular Binding

SwissADME [20] target was utilized to discover possible targets for the two molecules [21]. Four targets were detected, and more specifically, 5DYW for Butyrylcholinesterase, 4EY7 for Acetylcholinesterase, 5UEN for Adenosine A1 receptor, and 1YK8 for Cathepsin K. In all these crystal files the macromolecule was crystallized with a target. Molecular docking [22] has been applied for all targets. The grid parameters used were the same for all the substrates.: X = 40, Y = 40, Z = 40 (default) and the distance of the dots: 0.375 Å (default).

Then the coordinates from the co-crystallized ligand which were used for the active center of each macromolecule were: 5DYW [23]: X = 14.209, Y = 26.367, Z = −41.477 4EY7 [24]: X = −18,53, Y = −41.928, Z= 24.258, 5UEN [25]: X = 52.632, Y = 56.381, Z = 141.281, 1ΥΚ8 [26] X = 71.977, Y = 12.669, Z = 131.026

All docking scores are shown below in Table 6.

Regarding the 5UEN macromolecule, the binding energy values are the same for both compounds (ΔG = −7.46 kcal/mol). Thus, they strongly bind to Adenosine A1 receptors. The same results apply to the macromolecular 5DYW. The binding energy is −7.75 kcal/mol for thiocarbohydrazone **KKI18** and −8.24 kcal/mol for thiosemicarbazone **KKI15**. Thus, they strongly bind to butyrylcholinesterase.

Proceeding to the 4EY7 macromolecule, it was observed that both **KKI15** and **KKI18** bind strongly to acetylcholinesterase, with binding energy values of −8.6 and −9.08 kcal/mol, respectively.

In the latter macromolecule, the compounds do not bind strongly. Specifically, they gave values of −6.29 kcal/mol and −6.05 kcal/mol for **KKI15** and **KKI18,** respectively.

The in silico experiments are shown below. The binding energy and the inhibition constants were calculated computationally via the Autodock program.

Following the above conclusions, a thiosemicarbazone (**test2**) molecule and a thiocarbohdrazone (**test1**) molecule carrying an electronegative substituent in both rings were designed using the ChemDraw platform to control the influence of electronegativity on both rings and to examine the additive effect. The results are shown in Table 7 and Table 8 below.

In summary, both classes of compounds provide a strong binding at the macromolecules 5UEN and 4EY7. That is, the addition of electronegative substituents to both rings can also lead to compounds that bind strongly to these enzymes but with no additional binding energy. With respect to the 5DYW enzyme, the compounds are almost non-binding, so the addition of electronegative substituents to both rings attenuates the binding to butyrylcholinesterase. This may be attributed to the size of the compounds. In the latter macromolecule, the compounds do not bind strongly; a similar result was observed obtained with **KKI15** and **KKI18** compounds. The interactions of the **test1** compound with acetylcholinesterase differ from **KKI15**. It forms three hydrogen bonds with SER293 amino acid and two p-p interactions with aminoacid TRP286. Furthermore, **test2** compound forms one hydrogen bond with aminoacidARG296 in contrast with **KKI18** in that it does not form any bond with acetylcholinesterase. Secondly, in the adenosine A1 receptor, the test compounds form more interactions than **KKI15** and **KKI18**. All in all, the additions of substituents in three of the four macromolecules do not affect the binding energy. 

### 2.8. Results of the Pharmacokinetics and Toxicity Properties of the Two Compounds

Table 9 shows the Drug-Likeness Properties of **KKI15** and **KKI18.**

According to pkCSM and preADMET, thiosemicarbazone **KKI15** has a molecular weight < 500 g/mol, hydrogen bonding donors < 5, number of hydrogen bonding acceptors < 10, and lipophilicity less than five. As a result, it obeys Lipinski’s Rules of Five. Because lipophilicity is less than five, it can be easily absorbed from the body. Also, Veber’s Rule is qualified because the rotable bonds are less than seven. The same conclusions apply to thiocarbohydrazone **KKI18** (Table 10).

According to preADMET, the BBB value is less than one. As a result, **KKI15** is classified as inactive in Central Nervous System (CNS). In contrast, **KKI18** has a BBB value of more than one, and it might affect the Central Nervous System (CNS). The value for Human intestinal absorption is high for both compounds, and this signifies that these compounds might be better absorbed from the intestinal tract on oral administration. Moreover, **KKI18** was found to be an inhibitor of Pgp, and this means that it can increase oral bioavailability. According to pKCSM, both compounds might have the ability to penetrate the blood-brain barrier and thus be potential drugs for the Central Nervous System (CNS). Both compounds are not inhibitors of CP isoenzymes and therefore are not toxic or do not exert other unwanted adverse effects.

According to pKCSM, **KKI15** and **KKI18** have not been predicted to be hepatotoxic (Table 11). Moreover, AMES toxicity is negative, and that means that the compounds are not mutagenic. Finally, they are inhibitors of Herg II and, according to preADMET, present a low risk for Herg II.

## 3. Materials and Methods

### 3.1. Synthesis

Chalcone **A** was obtained using Claisen–Schmidt aldolic condensation and standard conditions [27]. Briefly, 55 mmoles of NaOH were dissolved in 95% EtOH/H_2_O 1:1 (250 mL), then 50 mmoles of acetophenone were added, and the mixture was cooled to 0 °C. Benzaldehyde, 50 mmoles, dissolved in 95% EtOH (50 mL), was added dropwise, and the reaction mixture was left to reach room temperature gradually for 24 h. The solid formed was filtered, washed with 95% EtOH/H_2_O 1:1, and used for the next preparations after drying in a desiccator over P_2_O_5_.

The corresponding thiosemicarbazone **KKI15** and thiocarbohydrazone **KKI18** were synthesized from chalcone **A**, which was subjected to a condensation reaction with thiosemicarbazide or thicarbohydrazide, respectively (Figure 1). The reactant ratio was a controlled condition to avoid the formation of bis-thiocarbohydrazone, and it gave the mono-adduct **KKI18** or **KKI15** exclusively.

#### Experimental

(*E*)-2-((*E*)-1,3-diphenylallylidene)hydrazine-1-carbothioamide, **KKI15**



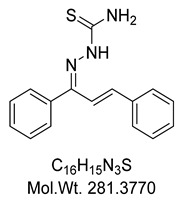



To a stirred solution of chalcone **A** (0.50 g, 2.40 mmoles) in 95% EtOH (60 mL), thiosemicarbazide (0.22 g, 2.40 mmoles) is added, followed by 3 drops of 6N HCl. The reaction mixture was stirred at room temperature for 24 h, then concentrated in vacuo, and the remaining was purified by column chromatography using PΕ(40–60)/AcOEt:8/2 as eluent to give thiosemicarbazone **KKI15** as yellowish solid in 66% yield (0.45 g), Rf = 0.73 (PE/AcOEt 8/2).

*N*’-((1*E*,2*E*)-1,3-diphenylallylidene)hydrazinecarbothiohydrazide, **KKI18**



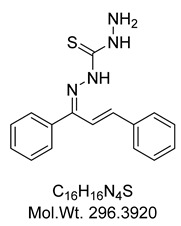



To a stirred solution of thiocarbohydrazide (0.20 g, 1.92 mmoles) in a mixture of 95% EtOH (58 mL) and H_2_O (20 mL), chalcone **A** (0.20 g, 0.96 mmoles) is added followed by 3 drops of 6N HCl. The reaction mixture was stirred at 80 °C for 3 h and then left to crystalize at room temperature for 24 h. The formed solid was filtered, washed with EtOH/H_2_O 1:1, and recrystallized from EtOH 95% to give **KKI18** in the form of a slightly ochre solid in 55% yield (0.31 g).

### 3.2. Structure Assignment

The two molecules under study have been structurally identified using an 800 MHz spectrometer (Bruker Avance spectrometer-Bruker Biospin GmbH, Reinsteten, Germany) installed in the NMR Centre in the National Institute of Chemistry in Slovenia using 1D and 2D homonuclear and heteronuclear experiments. The pulse sequences were obtained from the library of the spectrometer. The spectra were processed and analyzed using the MestreNova software.

### 3.3. Computational Details: DFT Calculations

Conformation analysis: Conformational analysis of the compounds has been carried out to find the lowest energy minimum structures employing the density functional theory (DFT) [28] at the B3LYP [29,30]/6-311 + G(d,p) [31] level of theory. This methodology is considered suitable for conformational analysis as it provides consistent results [32]. All conformers were fully optimized, and subsequently, their frequencies were computed in order to confirm that they are true minima. All calculations were carried out in DMSO solvent employing the polarizable continuum model (PCM) [33,34]. This model is divided into a solute part lying inside a cavity, surrounded by the solvent part represented as a material without structure, characterized by its macroscopic properties, i.e., dielectric constants and solvent radius. Furthermore, a solvent DMSO molecule has been added explicitly in all minimum structures and transitions states; see below. Finally, all theoretical data are compared with the experimental results obtained from the NMR spectra.

Activation Energy and transitions states: The paths of the isomerization process were calculated. The transition states connecting the various minima structures were computed using the synchronous transit-guided quasi-Newton (STQN) method [35].

Population analysis: The charges of the atoms were calculated using three population analyses, i.e., Mulliken, NBO and CM5 [36]. The B3LYP [28], M06-2X [37], ωB97-ΧD [38], and 3 basis sets, i.e., 6-311G(d,p) [without diffuse functions], 6-311 + G(d,p) [with diffuse functions for all atoms but H], and 6-311 ++ G(d,p) [with diffuse functions for all atoms]. All calculations were conducted using the Gaussian16 program [39].

Spectra calculations: UV-vis absorption and ECD spectra were calculated for the lowest forty excited states, using the TD-DFT methodology at the B3LYP/6-311 + G(d,p) level of theory. [40]

Circular Dichroism: Circular Dichroism was performed with a Jasco J-815model, featuring a peltier model PTC-423S/15 thermo stabilizing system. The cell used was 1 mm Quartz Suprasil cell. Concentrations used for the two samples were about 2 × 10^−4^ g/mL.

### 3.4. Molecular Binding

AutoDock [41] software was used for the molecular binding calculations and, more specifically, the Lamarckian Genetic algorithm. The crystal structures of the proteins were used by the online database “Protein Data Bank—PDB” and downloaded directly to the AutoDock program for study. The compounds used as ligands were designed with the help of the ChemOffice program, and using the same program, their energy was minimized with an MM2 force field.

### 3.5. ADMET Calculations

Both compounds were sketched in Chem Draw, and they were converted into SMILES in both web application tools preADMET [42] and pKCSm [43]. Both programs were used to examine the drug-likeness (Lipinski’s Rule of Five [44] and Veber’s rule [45]). Moreover, several toxicity parameters were examined via these tools. This procedure is very important for computational drug design because some potential biological compounds fail to reach the clinical trials because of their unfavorable (ADME) parameters.

## 4. Conclusions

This study focuses on structure assignment and conformational analysis of thiosemicarbazone adduct **KKI15** and thiocarbohydrazone **KKI18** using a combination of NMR spectroscopy and computational studies (QM methods).

The conformational analysis showed that both molecules could obtain four configurations, but the most favorable is when the two double bonds adopt the *E relationship*. Through 2D-NOESY and DFT, the conformations that are most probably to agree are MS1 and MS2 for **KKI15** and MS5 and MS6 for **KKI18**. The calculation of transition proves that the molecules obtain these two conformations.

In silico experiments were performed with four macromolecules to find some possible biological targets. The results showed that the compounds bind strongly to acetylcholinesterase, butyrylcholinesterase, and adenosine A1 receptors. In cathepsin K, they bind weakly. Both derivatives are not hepatoxic, and they obey Lipinski’s Rule of Five.

It appears that both molecules can be safe and bioactive, and putative for biological targets. This is very useful to synthetic chemists who wish to investigate new structures for various targets. The ultimate goal is through in silico molecular modeling screening to synthesize molecules that act selectively on certain biological targets.

## Data Availability

The data presented in this paper are available in Supporting information.

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
