# Peer review of "Conformational Properties of New Thiosemicarbazone and Thiocarbohydrazone Derivatives and Their Possible Targets"

_molecules, 2022, doi:10.3390/molecules27082537_

Round 1

Reviewer 1 Report

Authors are presenting a combined theoretical and experimental work on new thiosemicarbazone and thio-2 carbohydrazone derivatives. These compounds were synthetized and characterized. Their possible target enzymes were also identified. The work is of interest in order to find new class of drugs.

The presentation is very scholar. It looks like a practical student work.

Authors should rework the manuscript in order to make more concise. Several aspects should be developed: the results of the population analysis and its connection with the conformations found, ... Also they should perform in-depth analysis of their results and findings.

Author Response

We thank considerably the reviewer for his/her comments. The results were analyzed further and a depth analysis of our results was performed.

Reviewer 2 Report

The manuscript of Georgiou and collegues titled "Conformational properties of new thiosemicarbazone and thiocarbohydrazone derivatives and their possible target" reports the structural ssignment and conformational analysis of a thiosemicarbazone and its related thiocarbohydrazone by combining NMR and quantum mechanics calculations. In order to find a plausible biological application to the two molecules in silico studies were performed. Although the paper is strictly methodologic, it could help medicinal chemists working on thiosemicarbazones and analogues for a proper interpretation of the relationship between active conformation and biological activity of this class of molecules.

The manuscript is suitable for publication after addressing some issues.

  • In the introduction, the discussion on chalcones is too long and goes beyond the real discussion on thiosemicarbazones and thiocarbohydrazones. The part on chalcones should be shortened by focusing more on thiosemicarbazones.
  • The authors refer to the pharmaceutical potential of chalcones and thiosemicarbazones, generally reporting their use for the treatment of various pathologies. However, there is a lack of references to support these claims. The bibliography must therefore be implemented in agreement.
  • Precisely because both chalcones and thiosemicarbazones have been extensively studied from a pharmaceutical point of view and are able to interact with different biological targets, it can limit their real use as potential drugs. Accordingly, there are currently no drugs approved for use in humans with these facilities. This should be reported to warn the reader of the potential risk of working with these molecules 
  • The reason why the authors decided to extensively investigate the conformation of the two reported compounds (KKI15 and KKI18) should be specified. Why was it decided to examine these two molecules? Do they have any particular advantageous features?
  • To help the reader in the interpretation of the NMR signals of the two widely detailed compounds, it would be appropriate to report in the main text at least the proton spectra of KKI15 and KKI18 and to report the chemical shifts of the protons and related attributions in a table
  • In line 100 a certain compound NGI25 is mentioned. Is this a possible typo?
  • In figure 2 I would report for both compounds the conformations E and Z referred to during the discussion
  • In the header of table 1 it would be more appropriate to specify that it is the difference in energy between the various conformers and the conformer at the minimum energy, rather than simply writing Energy (kcal / mol)
  • Again in table 1, in the row referring to compound KKI18 there are numerical values, but it is not clear what they refer to
  • On lines 141-145 the authors state that "the structures of the compounds and the correlations that were observed in 2D-NOESY spectra's, the stuctures MS1 and MS5 are taken as the most probable for KKI15 and KI18." however, previously, the only description of NOESY interactions refers to the H9 and H7 protons of KKI15. Similarly for KKI18.
    To support the conformational results, it would be appropriate for the authors to highlight the effective NOE effect between the protons H7 and H3 identifying the MS1 conformation. The same for KKI18.
  • In figure 3 it would be appropriate to associate the representation of the conformation with the lewis formula to the ball-and-stick representation for a more immediate use by the reader.
  • The legends and units on the two axes of the graphs in figure 6 and 7 are missing
  • The paragraph on lines 175-180 is not too clear
  • Line 179 states that the energy gap between the comparisons MS5 and MS6 for the compound KKI18 is sufficiently low to guarantee the interconversion between the two conformers, as also demonstrated by the experimental data. However, for the compound KKI18 it is never referred to that in the NMR spectrum two pairs of signals that can be associated with the two conformers are seen. please specify better.
  • Table 5, 6, 7 and 8 can be unified in a single table.
  • Report, at least in SI, the images representing the binding mode and the protein-ligand complexes obtained from the docking studies.
  • In table 5-8 is reported an Inhibition constant. How it was calculated? Experimentally or computationally? If we are in the second case, how was it determined?
  • What is the reason that led the authors to design the two compounds test 1 and test 2? is there a reason why we have chosen to insert those substituents on the two aromatic rings?
  • Line 234-239: comment in more detail if the introduction of substituents in the compounds test 1 and test2 favors or disadvantages the binding with the targets investigated through an analysis of the binding mode and of the additional or missing interactions that these derivatives are able to form , as well as being based on the binding score.
  • Move the conclusion before material and methods
  • Line 327: why all the calculation were carried out in DMSO?

Author Response

We are submitting a revised version taken into account all the comments rose by the second reviewer. His/her comments are welcome and implemented in the revised version. The responses for every comment are in the uploaded pdf file.

Round 2

Reviewer 1 Report

Authors dis consider may concerns.

Reviewer 2 Report

I thank the authors for having found my suggestion appropriate for improving the quality of their manuscript, which is suitable for publication in the present form